



# Pairing litter decomposition with microbial community structures using the Tea Bag Index (TBI)

Anne Daebeler[1,2], Eva Petrová[1], Elena Kinz[3], Susanne Grausenburger[4], Helene Berthold[5], Taru Sandén[6], and Roey Angel[1,7] and High school students of biology project groups I, II, III from 2018-2019[+]

[1]Soil & Water Research Infrastructure (SoWa), Biology Centre CAS, České Budějovice, Czechia
[2]University of Vienna, Centre for Microbiology and Environmental Systems Science, Division of Microbial Ecology, Vienna, Austria
[3]Open Science - Life Sciences in Dialogue, Vienna, Austria
[4]Federal College for Viticulture and Fruit Growing, Klosterneuburg, Austria
[5]Institute for Seed and Propagating Material, Phytosanitary Service and Apiculture, Department for Seed Testing, Austrian Agency for Health and Food Safety (AGES), Vienna, Austria
[6]Institute for Sustainable Plant Production, Department for Soil Health and Plant Nutrition, Austrian Agency for Health and Food Safety (AGES), Vienna, Austria
[7]Institute of Soil Biology, Biology Centre CAS, České Budějovice, Czechia
[+]A full list of authors appears at the end of the paper

Correspondence to: Roey Angel (roey.angel@bc.cas.cz) or Taru Sanden (taru.sanden@ages.at)

Abstract. Including information about soil microbial communities into global decomposition models is critical for predicting and understanding how ecosystem functions may shift in response to global change. Here we combined a standardised litter bag method for estimating decomposition rates, Tea Bag Index (TBI), with high-throughput sequencing of the microbial communities colonising the plant litter in the bags. Together with students of the Federal College for Viticulture and Fruit Growing, Klosterneuburg, Austria, acting as citizen scientists, we used this approach to investigate the diversity of prokaryotes and fungi colonising recalcitrant (rooibos) and labile (green tea) plant litter buried in three different soil types and during four seasons with the aim of (i) comparing litter decomposition [decomposition rates (k) and stabilisation factors (S)] between soil types and seasons, (ii) comparing the microbial communities colonising labile and recalcitrant plant litter between soil types and seasons (iii) correlating microbial diversity and taxa relative abundance patterns of colonisers with litter decomposition rates (k) and stabilisation factors (S). Stabilisation factor (S), but not decomposition rate (k), correlated with the season and was significantly lower in the summer. This finding highlights the necessity to include colder seasons in the efforts of determining decomposition dynamics in order to quantify nutrient cycling in soils accurately. With our approach, we further showed selective colonisation of plant litter by fungal and prokaryotic taxa sourced from the soil. The community structures of these microbial colonisers differed most profoundly between summer and winter, and rooibos litter was generally a stronger selector than green tea litter. Moreover, this study indicates an equal, if not higher, importance of fungal vs prokaryotic degraders for recalcitrant and labile plant litter decomposition. Our results collectively demonstrate the importance of analysing decomposition dynamics over multiple seasons and isolating the effect of the active component of the microbial community.



## Introduction

Litter decomposition is one of the most important terrestrial ecosystem functions. Soil microorganisms drive this process by breaking down plant material, leading to the release of carbon into the atmosphere as $CO_2$ or into the soil, where it either gets sequestered or further degraded (Talbot and Treseder, 2011; Allison et al., 2013; Cotrufo et al., 2013). Litter decomposition, therefore, directly affects both Earth's atmosphere and soil health. Understanding this process holds the key to better agricultural management, mitigating greenhouse gas emissions, and predicting future soil carbon storage levels. However, because of the processes' complexity and our inadequate understanding of how biodiversity affects it, applying the litter decomposition process to agricultural practices and global decomposition models currently does not perform at its full potential. Soil microbial community structure has been hypothesised to influence process rates in soils (McGuire and Treseder, 2010; Graham et al., 2016). In a few models that linked microbial diversity and decomposition (e.g., MIMICS, (Wieder et al., 2014) and MEMS (Cotrufo et al., 2013)), microbial diversity positively affected nutrient-cycling efficiency and ecosystem processes through either greater intensity of microbial exploitation of organic matter or functional niche complementarity.

Litter quality has been shown to be important for microbial community structure since both bacteria and fungi respond to litter physicochemical changes during the decay process (Aneja et al., 2006; Purahong et al., 2016). Litter biochemical traits, such as the C/N ratio and the fraction of lignin, are considered good indicators of litter quality, as they are related to nutrient availability and decomposition stage (Prescott, 2010; Talbot and Treseder, 2012). Different litter types can thus select microbial taxa that are more specialised in degrading their components. Fungi are known to produce a suite of oxidative enzymes that degrade the recalcitrant biopolymers of litter (Mathieu et al., 2013; Hoppe et al., 2015). In contrast, only a few types of bacteria degrade all lignocellulosic polymers, while most typically target simple soluble compounds (de Boer and van der Wal, 2008). Therefore, the role of bacteria in the decomposition of more recalcitrant material is still debated (Wilhelm et al., 2018). However, still little is known on how microbial communities specialise in litter types with different physical and chemical traits (Freschet et al., 2011; Pioli et al., 2020).

The community structure and functioning of microbial communities are further affected by biotic interactions (Daebeler et al., 2014; Garbeva et al., 2014; Ho et al., 2016; Wilpiszeski et al., 2019). In natural communities, such interactions generally involve competition for space and resources (Boddy, 2000; Faust and Raes, 2012) and references therein), but may also be mutualistic. For example, it has been suggested that bacteria can facilitate the activity of decaying fungi by providing important nutrients such as nitrogen (N) and phosphorous (P) (Purahong et al., 2016). Likewise, fungi have been demonstrated to help improve the accessibility of litter for bacteria (de Boer et al., 2005). Therefore, it is likely that decomposition dynamics depend not only on microbial community structure but also on the facilitative or competitive interactions among microbial species (Liang et al., 2017; Aleklett et al., 2021). Despite the indications of the high importance of species interactions for soil organic matter decomposition dynamics, they are not well understood, and more studies under natural conditions are needed.

Linking microbial diversity to function across different environments represents a key aspect of ecology. In this study, we used high-throughput sequencing in combination with the Tea Bag Index (TBI)-a cost-effective method to study litter decomposition using commercially available tea bags (Keuskamp et al., 2013). This combination of methods allows gaining insight into the effect of litter traits on the community structure of microbial decomposers, as well as linking soil organic matter decomposition rates with microbial population dynamics. The microbial diversity in soils is immense (Thompson et al., 2017), yet large parts of it are dormant (Lennon and Jones, 2011) or simply do not participate in litter degradation (Falkowski et al., 2008). Therefore, the buried teabags used in this study serve as "traps" for microbial





litter degraders and assure that the diversity we observe is composed only of its active litter degrading and associated taxa.

Specifically, in this citizen-science-aided project, we used barcoded-amplicon high-throughput sequencing of the SSU rRNA gene (for bacteria and archaea) and the internal transcribed spacer region (ITS) for fungi to compare the microbial community structure in two different standardised litter types (rooibos and green tea) with the microbial community in

three different local soil types and during four seasons. We tested the extent to which litter types with different traits represent selective substrates for microbial community colonisation. Finally, we related the microbial diversity and species relative abundance patterns with two proxies (decomposition rate (k) and stabilisation factor (S)) that describe the decomposition of labile material. We hypothesised that i) microbial diversity in the tea bags will correlate positively with decomposition rates and negatively with stabilisation factors, and ii) different subsets of local soil microbiota will colonise

labile and more recalcitrant litters, and thus each litter type will select for a different, specialised community. To the best of our knowledge, this is the first study utilising the Tea Bag Index method in combination with a microbiome analysis to investigate how litter decomposition is linked with microbial community structure and diversity.

## Materials and methods

### Description of the study sites

The study site is located at the Agneshof vineyards of the agricultural college for Viticulture and Pomology in Klosterneuburg, just north of Vienna, Austria (Fig. 1). The mean annual temperature was 11.1 °C, and the mean total annual precipitation was 689 mm between 2010 and 2019 at the Agneshof weather station. We selected three study plots with three different soil types: Fluvisol, Cambisol, and Luvisol to maximise the difference in soil characteristics. The Fluvisol and Luvisol (Alfisol) sites were cultivated with wine, whereas the Cambisol site was a set-aside grassland

between vineyards. At Fluvisol and Luvisol sites, we selected one inter-row sampling area in the middle of the vineyard at least 5 m distance to the vineyard edge. In contrast, at the Cambisol, we selected a sampling area that was at least 5 m distance to the neighbouring vineyard.

### Soil sampling and soil chemical analyses

Composite soil samples of 10-12 individual soil cores were taken from 0-10 cm depth in one to three field replicates every

three months between March 2018 and December 2018 to characterise the sites. Soils were sieved through a 2 mm stainless sieve and air-dried prior to further analyses. Soil pH was measured electrochemically (pH/mV Pocket Meter pH 340i, WTW) in 0.01 M $CaCl_2$ at a soil-to-solution ratio of 1:5 (ÖNORM L1083). Plant available phosphorous (P) and potassium (K) were determined by calcium-acetate-lactate (CAL) extraction (ÖNORM L1087). Total soil organic C concentrations of the soil samples were analysed by dry combustion in a LECO RC-612 TruMac CN at 650 °C (ÖNORM

L1080; LECO Corp.). Total N was determined according to ÖNORM L1095 with elemental analysis using a CNS (carbon, nitrogen, sulfur) 2000 SGA-410–06 at 1250 °C. $KMnO_4$ determination of labile carbon was analysed according to Tatzber et al. (2015). Potential nitrogen mineralisation was measured by the anaerobic incubation method (Keeney, 1982), as modified according to Kandeler (1993). The soil texture was determined according to ÖNORM L1061-1 and L1062-2.

### Tea Bag Index (TBI)

The Tea Bag Index [i.e., the decomposition rate (k) and stabilisation factor (S)] was assessed according to (Keuskamp et al., 2013) between December 2017 and December 2018. In short, commercially available green tea (EAN 87 22700 05552



5) and rooibos tea (EAN 87 22700 18843 8) in non-woven, polypropylene mesh bags produced by Lipton (Unilever) were used as standardised litter bags. Green and rooibos teabags in eight replicate pairs (four replicate pairs for calculating the TBI parameters and four replicate pairs for molecular analysis) were weighed and buried pairwise at a depth of 8 cm for

three months (Winter: December 2017-March 2018, Spring: March 2018-June 2018, Summer: June 2018-September 2018, Fall: September 2018-December 2018). There was a 15 cm distance between the green and rooibos tea bags within a replicate pair and at least 75 cm between the replicate pairs. Subsequently, the teabags used for calculating the TBI were retrieved, cleaned of adhering soil particles, and dried for at least three days on a warm, dry location before re-weighing. Values for k and S were calculated using the mass losses of green and rooibos teas, as described in (Keuskamp et al.,

2013). The teabags used for molecular analysis were frozen on-site on dry ice after being dug out and cleaned and transported in a frozen state to the Anaerobic and Molecular Microbiology lab at SoWa, BC CAS, České Budějovice, Czechia, for further analysis. In addition, 1-3 composite soil samples from the vicinity of the burying place of the teabags were collected at each season for comparison.

Molecular analyses of prokaryotic and fungal community structures

DNA extraction and amplicon sequencing

DNA was extracted from 0.25 g of frozen tea material or soil using the DNeasy PowerSoil DNA Isolation Kit (Qiagen) according to the manufacturer's instructions. Amplicon sequencing was done using a two-step barcoding approach (Naqib et al., 2019). DNA was also extracted from two unburied green tea and rooibos teabags to estimate the microbial load and diversity already present on the tea material. The DNA extracts were then quantified using Quant-iT™ PicoGreen™

dsDNA Assay Kit (Thermo) and diluted to 10 ng µl$^{-1}$ for use as templates for PCR amplification. The V4 region of the 16S rRNA gene (16S hereafter) was amplified using primers 515F-mod-CS1 (aca ctg acg aca tgg ttc tac aGT GYC AGC MGC CGC GGT AA), 806-mod-CS2 (tacggtagcagagacttggtctGGACTACNVGGGTWTCTAAT; Walters et al., 2015) while ITS was amplified using primers ITS1f-CS1 (acactgacgacatggttctacaCTTGGTCATTTAGAGGAAGTAA) and ITS2-CS2 (tacggtagcagagacttggtctGCTGCGTTCTTCATCGATGC; Gardes and Bruns, 1993). Amplification was done

in a T100 Thermal Cycler (Biorad) with amplification cycles ranging from 25 to 28 for 16S and 29 to 32 cycles for ITS, depending on the amount of template. The full protocol is available online (Angel, 2021). Before processing the samples from the experiment, we performed a preliminary test using the chloroplast blocker pPNA (PNA Bio). For this purpose, DNA from the unburied tea bags, from two buried teabags and a soil sample was used for amplification with or without PNA (0.25 µM, final conc.). If PNA was used, an additional PCR step of PNA clamping (75 ºC for 10 s) was added in

each cycle between the denaturation and primer annealing step, according to the manufacturer's instructions. In addition, no-template PCR controls (NTC) and "blank extraction control" (DNA extraction and amplification without a sample) were sequenced in each batch (season). A mock community (ZymoBIOMICS Microbial Community DNA Standard II; Zymo Research) was also amplified and sequenced. Library construction and sequencing were performed at the DNA Services Facility at the University of Illinois, Chicago, using an Illumina MiniSeq sequencer (Illumina) in the 2 × 250

cycle configuration (V2 reagent kit).

Sequence data processing and classification

Primer regions were trimmed off the amplicon sequence data using cutadapt (V2.3; Martin, 2011). Downstream sequencing processing steps were done in R (V4.0.3; R Core Team, 2020). Quality-trimming and clustering into amplicon sequence variants (ASVs) were done using the DADA2 pipeline (Callahan et al., 2016). For 16S, the

following quality filtering options were used: `no truncate, maxN = 0, maxEE = c(2, 2)`



and truncQ = 2. For ITS, the following options were used: minLen=50, maxN = 0, maxEE = c(2, 2) and truncQ = 2. Chimera sequences were removed with `removeBimeraDenovo()` using the "consensus" method "allowOneOff". Taxonomic classification of the 16S ASVs was done with `assignTaxonomy()` against the SILVA database (Ref NR 99; V138.1; Quast et al., 2012), while for ITS, it was done against the UNTIE

database (Nilsson et al., 2018). Potential contaminant ASVs were removed using `decontam` (Davis et al., 2017), employing the default options. Unclassified taxa and those classified as either "Eukaryota", "Chloroplast", or "Mitochondria" (in the 16S dataset) or as "Bacteria" or "Archaea" (in the ITS dataset) were removed. In addition, for beta-diversity analysis ASVs, appearing in <5% of the samples were removed.

Statistical analysis

All statistical analysis was performed in R (V4.0.3; R Core Team, 2020). Significant differences between the decomposition rates (k) and stabilisation factors (S) at the three study sites during the four studied seasons, was determined using Tukey's HSD on the estimated marginal means (Lenth, 2021); functions `emmeans()` and `contrast()`). Spearman rank ($\rho$) correlations were performed to investigate the relationship between soil properties and the decomposition rates, stabilisation factors, and microbial amplicon sequence variant (ASV) richness

and diversity. Sequence data handling and manipulation were done using the `phyloseq` package (McMurdie and Holmes, 2013). For alpha-diversity analysis, all samples were subsampled (rarefied) to the minimum sample size using a bootstrap subsampling with 1000 iterations to account for library size differences, while for beta-diversity analysis, library size normalisation was done by converting the data to relative abundance and multiplying by median sequencing depth. The Inverse Simpson's, Shannon's H diversity indices and the Berger-Parker dominance index were calculated using the

function `EstimateR()` in the `vegan` package (Oksanen et al., 2018) and tested using ANOVA in the stats package, followed by a post hoc Tukey's HSD test on the estimated marginal means. Variance partitioning and testing were done using PERMANOVA (Mcardle and Anderson, 2001) function `adonis()` using Horn-Morisita distances (Horn, 1966). Differences in order composition between the sample types were tested similarly to STAMP (Parks et al., 2014), but written in R. Briefly, the relative abundance of the orders was compared between samples using the Kruskal-Wallis rank-

sum test (`kruskal.test()` in the `stats` package), followed by the Mann–Whitney post-hoc test (`wilcox.test()` in the `stats` package), and FDR corrected using the Benjamini–Hochberg method (Benjamini and Hochberg, 1995); `p.adjust()` in the `stats` package). Differentially abundant ASVs were detected using ALDEx2 (Fernandes et al., 2013) using the option `denom = iqlr` for the `aldex.clr()` function. Differences in composition between the two teabag types and the soil were tested in a pairwise manner. Plots were generated using `ggplot2` (Wickham, 2009).

Results and discussion

Soil chemical characteristics and Tea Bag Index

To be able to draw conclusions about the connections between the community structures of microbial litter decomposers, litter types and the decomposition process, we first determined the basic soil characteristics of the three study plots. The Luvisol plot exhibited the lowest contents of P, K, TOC, sand, and the highest pH (Table 1).

Next, we determined the decomposition rate, k (which reflects the rate by which the labile fraction of the litter is decomposed ), and the stabilisation factor, S (which reflects the proportion of litter that breaks down), for all three study plots and during four seasons. Despite decomposition rates ranging from 0.008 g d$^{-1}$ at the Cambisol site to 0.025 g d$^{-1}$ at the Fluvisol site (in the summer), no significant differences could be found between sites or seasons at each site (Fig. 2.





The results are well within the range of summer data previously reported from Austria (Buchholz et al., 2017; Keuskamp et al., 2013; Sandén et al., 2020)(Sandén et al., 2021). However, to the best of our knowledge, we are the first ones to report data during four seasons. There was also no correlation of any measured soil properties with the decomposition rates (data not shown), as also reported in (Sandén et al., 2020). These results were surprising, given the differences in the determined soil characteristics (Table 1) and temperature between the seasons.

We did, however, observe significantly lower stabilisation factors in summer than in all other seasons ($p_{adjusted} \leq 0.001$). Specifically, the stabilisation factors determined at the Cambisol and Luvisol sites were significantly lower in summer than those determined for the other three seasons ($p_{adjusted} \leq 0.05$). These results indicate that the conditions during the summer months favour the decomposition of a larger fraction of the organic material, but not necessarily in a faster manner. This is in apparent contrast to the well-known relationship between temperature and decomposition rates (see Kirschbaum, 1995 and references therein). However, one should consider that: 1. temperature sensitivity of decomposition rate is much higher at low temp. 2. that the decomposition rate is also heavily affected by moisture which could counterbalance the effect of temperature. Similarly, (Elumeeva et al., 2018) also found a correlation of S but not of k to various edaphic factors. Notably, soil moisture, pH and altitude did not affect k (but were correlated with S). Since S measures "how much" (rather than "how fast") complex organic matter decomposes, it can be assumed that S is more sensitive than k to the composition of microbial communities that are active at the time of decomposition. This is because the underlying process of S involves the concerted (discrete) action of several microbial guilds, each specialising in decomposing one or more substrates sequentially (Zheng et al., 2021; Glassman et al., 2018). Temperature and soil moisture (and hence seasonality) have been demonstrated to affect litter decomposition dynamics (Prescott, 2010).

DNA extraction and sequencing statistics

Before investigating the microbial communities involved in the litter degradation measured by the Tea Bag Index (TBI) method, we extracted DNA from unburied teabags serving as negative controls and performed rRNA gene amplicon sequencing (16S and ITS regions). DNA yields were similar between buried and the unburied teabags and averaged $6.1 \pm 12.9$ and $23.0 \pm 14.0$ mg g$^{-1}$, probably because, as expected, most of the DNA originated from the plant cells (data not shown). A preliminary study using the chloroplast blocker pPNA was carried out to estimate the effect of plant chloroplast on the sequencing output (what proportion of the sequences are dominated by chloroplast). As Table S1 sows, only the unburied tea material had a significant portion of chloroplast reads (64.5% and 15.4% for green tea and rooibos, respectively), while after three months in the ground, no chloroplast reads were detected (most likely due to a combination of microbial colonisation and chloroplast DNA degradation). Using the pPNA blocker nearly eliminated the number of chloroplast reads while at the same time not biasing the community composition in the samples (Fig. S1). Nevertheless, since chloroplasts were undetected in the buried teabags, pPNA was not used in subsequent sample processing. Fig. S1 also shows that, as expected, the microbial load on the unburied tea material was minimal and yielded only about 5% of the reads after quality filtering. Such a low microbial load is expected for above-ground plant parts (e.g. Chen et al., 2020; Knorr et al., 2019).

To then study the prokaryotic and fungal communities potentially involved in litter degradation, we extracted DNA from soil and triplicate rooibos and green tea teabags buried for three months for each season at each site.

Richness and diversity of prokaryotic and fungal communities in tea bags and soils

Sequencing the prokaryotic 16S rRNA gene and the fungal ITS region and downstream data processing yielded a total of 16,100 16S ASVs and 3,685 ITS ASVs. After removing all ASVs with a prevalence of <5% of the samples, 4217 16S



ASVs and 402 ITS ASVs remained. As expected, archaea were rare and comprised only about 0.9% of the reads. At all three sites, the observed prokaryotic and fungal ASV richness (i.e., the observed number of prokaryotic and fungal ASVs) was roughly twice as high in the soil samples than in teabag samples (Fig. S2). Likewise, the microbial diversity, as estimated by the Inverse Simpson's, Shannon and Berger-Parker indexes, was generally significantly higher in the soil samples than in the teabag samples. There was, however, no significant difference between rooibos and green tea bags in terms of prokaryotic and fungal ASV richness or diversity. These findings indicate selective microbial colonisation of both the labile and the more recalcitrant litter sourced from the surrounding soil during all seasons.

Both the Fluvisol and the Cambisol site teabags buried in summer harboured a significantly richer prokaryotic community than those buried in other seasons. Such seasonal differences were not observed with the fungal communities. Similar to the observed differences in richness, the biodiversity of prokaryotic communities detected in the teabags was often significantly higher in summer samples than in all other seasons. This hints that there are more distinct prokaryotic species capable of colonising and degrading litter in summer than in the other seasons. An explanation for this observation could be the activation of decomposing prokaryotic soil populations by summer conditions such as elevated temperature (Kirschbaum, 1995; Allison et al., 2010). Finally, the prokaryotic ASV richness and diversity of the communities that colonised the teabags were negatively correlated with the stabilization factor S ($p < 0.01$; correlation factor $\rho = -0.60$ and $p < 0.01$; correlation factor $\rho = -0.54$, respectively). These findings suggest that less inhibition of litter decomposition occurred in the summer season, in which richer and more diverse prokaryotic communities were involved in the degradation. With these data, we can partially confirm our hypothesis of a positive correlation between microbial diversity and litter decomposition. Possibly, a positive relationship between microbial diversity and litter decomposition is prevalent in many soil types, as it has also been experimentally established in a Cambisol (Maron et al., 2018).

The richness and diversity of fungal ASVs were generally comparable across the seasons at all sites, except for a higher richness at the Cambisol site in winter compared to the other seasons. This could be explained by the hyphae growth form of many fungi, which allow them to span various microsites in the heterogeneous soil environment and make them less sensitive to chemical gradients and seasonal changes (Yuste et al., 2010).

Selective colonisation of tea bags by prokaryotic and fungal soil populations

After three months of burial, the community structures of prokaryotes and fungi detected in tea bags were significantly different from those detected in soil (Fig. S3). Moreover, the differential abundance analysis clearly showed enrichment of about a third of the 16S ASVs and 14-20% of the ITS ASVs in the teabags compared to the soil (Fig. S4). Therefore, as has been observed before in various soil systems and with a range of different litter types (Bray et al., 2012; Yan et al., 2018; Wei et al., 2020; Aneja et al., 2006), we conclude that the litter material in the teabags selected for colonisation by a minority portion of the bacterial, archaeal and fungal population acting as the active litter decomposers and associated populations. Comparing the green tea with the rooibos samples, we could detect a subgroup of ca. 4.2% and 2.5% of the 16S rRNA gene ASVs that preferentially colonised either the green tea or rooibos teabags. In contrast, only four ITS ASVs differed between the green tea and rooibos samples, indicating non-preferential colonisation. For both prokaryotes and fungi, sample type (i.e., soil green tea or rooibos) explained the largest fraction of the variance in the community composition (38 and 23%, respectively; Table S2). Although, not surprisingly, this was driven mainly by the differences between soil and teabag communities. However, the difference between green tea and rooibos was not negligible and explained 13 and 9% of the variance in a pairwise comparison for prokaryotes and fungi, respectively (Table S2, Fig. S3). The season was also a major factor, explaining 20 and 22% of the variance, respectively, and driven mostly by the difference between summer and winter (27 and 29% of the variance, respectively, Table S2, Fig. S4). In contrast, soil



type had only little effect, explaining only 3 and 5% of the variance, respectively. This is in contrast to the recent findings of (Pioli et al., 2020), in which a large effect of soil type on the community composition was reported, though not surprising considering that this study was performed on a very local scale.

Further interested in the identity and possible colonisation preferences of the microbial decomposers, we compared the relative abundance of dominant orders between the different teabags, the soil types and between the teabag communities in the summer vs winter, in a pairwise manner. This analysis allowed us to identify specific bacterial and fungal lineages that exhibited distinct colonisation patterns and were enriched in samples from the more labile, green tea or the more recalcitrant rooibos bags. Most strikingly, members of the Pseudomonadales, Sphingobacteriales, were the most enriched in both green tea and rooibos samples as compared to the soil samples from the same sites with differences of 11.4% and 9.4% for Pseudomonadales, and 8.2% and 8.3% for Sphingobacteriales (Fig. 3a, c). Additionally, green tea samples harboured 7.6% and 6.7% more Flavobacteriales and Micrococcales, respectively, than the accompanying soil samples (Fig. 3a). Rooibos teabags, on the other hand, were significantly richer in Betaproteobacterials by 4.4% and in Rhizobiales by 4.2% (Fig. 3c). Similar increases in relative abundance of Pseudomonadales, Flavobacteriales, Sphingobacteriales, Micrococcales have been observed in straw decomposing soil microcosms (Jiménez et al., 2014; Guo et al., 2020). Members of Pseudomonadales, Flavobacteriales and Rhizobiales are well known to have the capacity to degrade plant lignin, (hemi-)cellulose or carboxymethyl cellulose (Koga et al., 1999; McBride et al., 2009; Wang et al., 2013; Talia et al., 2012; Jackson et al., 2017). Furthermore, Pseudomonadales and Flavobacteriales can be involved in the degradation of furanic compounds (Jiménez et al., 2013; López et al., 2004). The increased presence of Sphingobacteriales may be explained by their capability to produce β-glucosidases that remove the cello-oligosaccharides produced by polymer degraders (Matsuyama et al., 2008) and is considered a critical and rate-limiting step in cellulose degradation (du Plessis et al., 2009). Indeed, soils with high β-glucosidase activity were also found to be dominated by members of the Sphingobacteriales (particularly from the Chitinophagaceae family; Bailey et al., 2013).

We further observed significant enrichments of fungal orders in both teabag types. Members of the Hypocreales had a 20% higher relative abundance in green tea than in soil but were 7% depleted in the rooibos compared to the soil (Fig. 3b). In contrast, Helotiales were enriched in the rooibos samples by 12% (Fig. 3d). Hypocreales are common saprophytic fungi and are known to produce cellobiohydrolases and endoglucanases necessary for the depolymerisation of celluloses (Lynd et al., 2002; Martinez et al., 2008). Helotiales are abundant, cellulolytic soil fungi that have been shown to specialise in degrading recalcitrant organic carbon (Newsham et al., 2018; Thomasgbif, 2020). This can explain their prevalence in the rooibos compared to the green tea samples.

An indication of a preference for the decomposition of more labile litter, or for an association with degraders of such material, is the higher relative abundance of the bacterial orders Pseudomonadales and Micrococcales in green tea than in rooibos samples by 2.8% and 3.6%, respectively (Fig. 3e) and of the fungal Hypocreales by 26.6% (Fig. 3f). On the other hand, in the more recalcitrant rooibos samples, Rhizobiales species were significantly more abundant by 5.1% (Fig. 3e). These findings support our hypothesis of a selection for specific microbial species by the different litter present in green tea and rooibos. Since the green tea litter contains a higher hydrolysable fraction and a lower C:N ratio (Keuskamp et al., 2013), we assume the Pseudomonadales and Micrococcales, which were significantly enriched in green tea samples, were involved in the degradation of more labile compounds such as cellulose.

The selection of microbial degraders colonising the litter inside the teabags resulted in a stronger relative enrichment of fungal than bacterial orders. Furthermore, fungal community structures showed a stronger response than bacterial

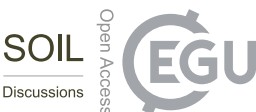

communities to the different litter types, as also observed by (Habtewold et al., 2020). Our results, therefore, indicate an equal (if not higher) importance of fungi for the degradation of both labile and recalcitrant litter in the studied soils.

## Conclusions

In contrast to our hypothesis, the different seasons did not affect the litter decomposition rate, but they did differ in how much organic carbon could be degraded. At the same time, microbial richness and diversity of litter-colonising degraders were positively correlated with the fraction of degraded organic carbon, indicating a positive relationship between litter degradation and microbial biodiversity in soil. Microbial colonisation of the tea litter was substrate-selective and season-dependent for prokaryotes and fungi. On average, about 28-30% of the prokaryotic ASVs and 14-19% of the fungal ASVs

preferentially colonised the rooibos and green tea, respectively. However, their exact identity varied between seasons, especially between the summer and winter. This work demonstrates the power of microbiome-resolved-TBI to give a holistic description of the litter decomposition process in soils.

## Code availability

The scripts to reproduce this and all downstream statistical analysis steps are available online
under `https://github.com/roey-angel/TeaTime4Schools`.

## Data availability

Raw sequence reads have been deposited in the Sequence Read Archive under BioProject accession no. PRJNA765214.

## Author contributions

TS, RA, and AD designed the study. TS, HB and EK performed sampling with support from students of biology projects
groups I, II, and III. EK developed lab protocols, EP optimised the lab protocols and performed the DNA extraction and PCR amplification. TS, RA, and AD analysed the data. The manuscript was written by AD, TS, and RA with contributions from all co-authors. We thank Michael Schwarz for preparing the map in Figure 1.

## Team list

Group I: Fabian Bauer, Lorenz Baumgartner, Lisa Brandl, Michael Buchmayer, Karl Daschl, Rene Dietrich, Stefan
Ebner, Margarete Jäger, Michaela Kiss, Lea Kneissl, Katja Langmann, Elias Mauerhofer, Jakob Paschinger, Tobias Rabl, Sophie Schachenhuber, Josef Schmid, Marlene Steinbatz. Group II: Mathias Ettenauer, Lorenz Hauck, Mathias Herl, Johannes Honsig, Thomas Gangl, Anja Glaser, Gabriel Hümer, Georg Lenzatti, Stefan Lichtscheidl, Lena Mayer, Valentin Oppenauer, Jacob Rennhofer, Katharina Schönner, Ciara Seywald. Group III: Jan Gamzunov, Anna Hess, Christoph Schurm, Klaus Stacher, Lena Ungersböck, Florian Valachovich, Mirjam Weissmann, Marius Wittek, Jennifer
Wosak, Valentin Zahel, Lucas Züger.



Financial support

This work was supported by the Austrian Ministry of Education, Science and Research (BMBWF) through a Sparkling Science Grant to TS, HB, EK, AD, RA and SG (SPA 06/044 TeaTime4Schools). In addition, EP and RA were supported
by the Czech MEYS (EF16_013/0001782 - SoWa Ecosystems Research). AD was supported by the FWF grant T938.





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





Tables and Figures

Table 1: Soil characteristics at the three study plots over one year.

| Plot | Season | pH | P | K | TOC | $N_{tot}$ | Labile C | Pot. mineralis. N | Sand | Silt | Clay |
|---|---|---|---|---|---|---|---|---|---|---|---|
| | | | (mg kg soil$^{-1}$) | (mg kg soil$^{-1}$) | (%) | (%) | (mg kg dry soil$^{-1}$) | (mg kg soil$^{-1}$ 7 days$^{-1}$) | (%) | (%) | (%) |
| Fluvisol | Winter | 7.17[a1] | 155[a] | 474[a] | 1.92[a] | 0.2[a] | 650[a] | 54.21[a] | NA | NA | NA |
| Fluvisol | Spring | 7.22 | 141 | 578 | 2.34 | 0.23 | 744 | 126.02 | NA | NA | NA |
| Fluvisol | Summer | 7.15 | 145 | 570.33 | 2.56 | 0.24 | 745.67 | 86.26 | 34.47 | 38.87 | 27.53 |
| Fluvisol | Autumn | 7.37 | 155.67 | 518.33 | 2.24 | 0.21 | 774.67 | 58.92 | NA | NA | NA |
| Cambisol | Winter | 7.07[ab] | 203[a] | 696[b] | 2.67[b] | 0.24[b] | 776[b] | 164.95[a] | NA | NA | NA |
| Cambisol | Spring | 7.17 | 191 | 624 | 2.52 | 0.25 | 774 | 185.23 | NA | NA | NA |
| Cambisol | Summer | 7.14 | 195 | 619 | 2.43 | 0.23 | 715 | 118.04 | 46.53 | 33.13 | 20.2 |
| Cambisol | Autumn | 7.24 | 188.33 | 561.33 | 2.19 | 0.21 | 706 | 78.71 | NA | NA | NA |
| Luvisol | Winter | 7.27[b] | 143[b] | 323[c] | 1.71[b] | 0.17[b] | 582[b] | 104.08[a] | NA | NA | NA |
| Luvisol | Spring | 7.39 | 108 | 282 | 1.65 | 0.16 | 550 | 107.36 | NA | NA | NA |
| Luvisol | Summer | 7.33 | 137 | 279.67 | 1.58 | 0.15 | 536.33 | 78.75 | 21.17 | 59.83 | 19.7 |
| Luvisol | Autumn | 7.44 | 136.33 | 329.33 | 1.81 | 0.17 | 613.33 | 92.94 | NA | NA | NA |

[1] Lowercase letters indicate significant differences between plots (adjusted $p \leq 0.05$)




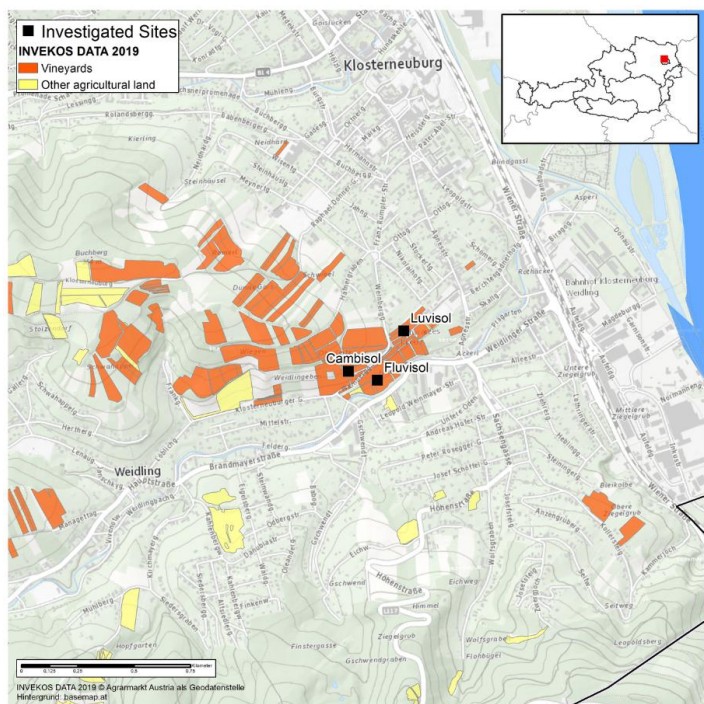

Figure 1: Geographical map of the investigated sites in Klosterneuburg, Austria. The study sites are marked as black squares.





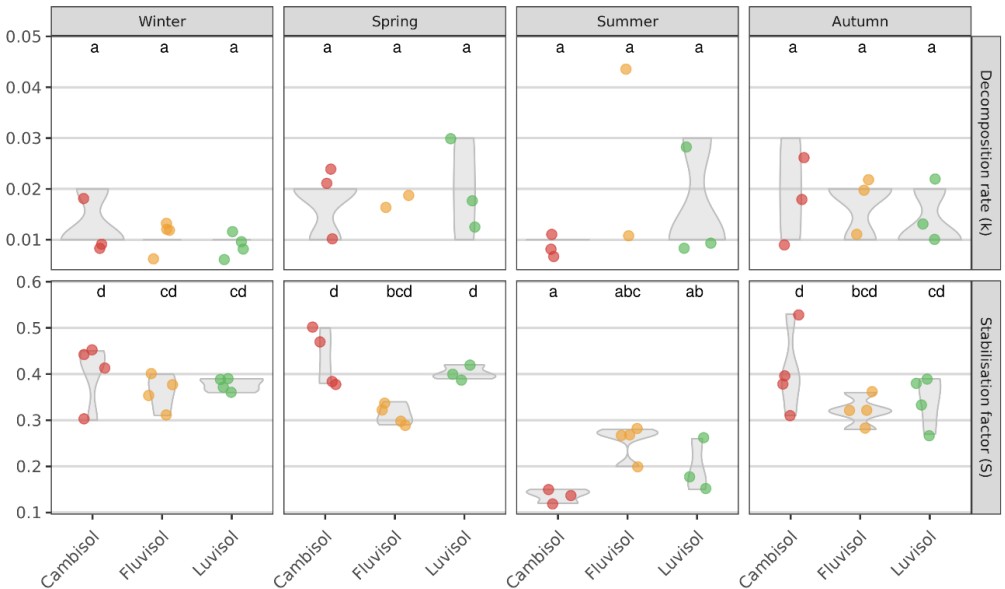

Figure 2: Decomposition rate (k) and stabilisation factor (S) determined at three sites with different soil types during four
seasons. Values shown are averages (n=4, lower in case a teabag broke during incubation or in case the rooibos was not in the
first phase of decomposition anymore and k could not be calculated) and the interquartile range (Violin shapes). Lowercase
letters indicate significant differences between samples (seasons and sites together; $p_{adj.} \leq 0.05$).



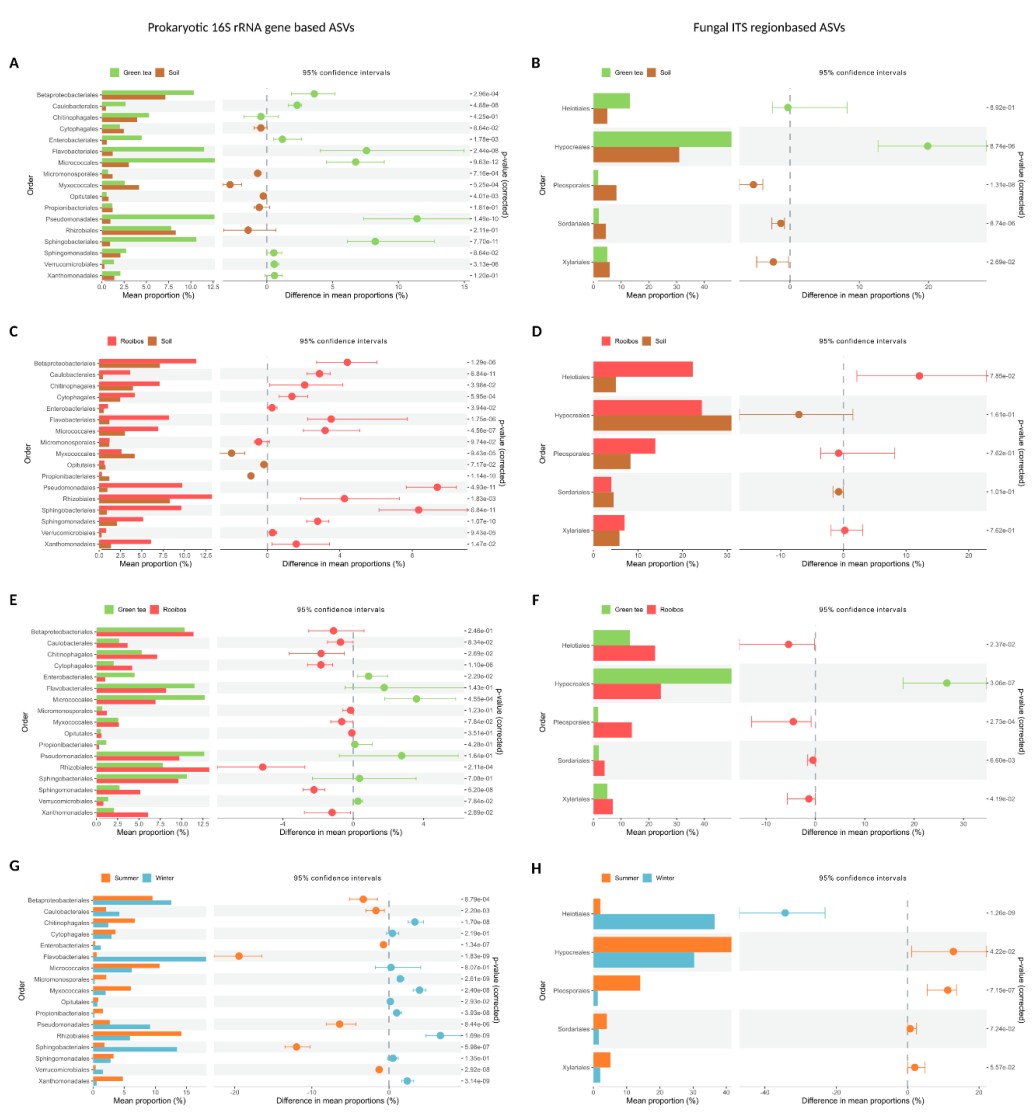

Figure 3: Comparative analysis of the relative abundance based on bacterial and fungal order profiles between sample types
and seasons. Samples were compared using Kruskal–Wallis rank-sum test, followed by the Mann–Whitney post-hoc test, and
FDR corrected using the Benjamini–Hochberg method. Only orders with a median relative abundance of >5% are shown.