# Peer review of "Pairing litter decomposition with microbial community structures using the Tea Bag Index (TBI)"

_SOIL, 2021_

## Referee Comment (RC1)

**General comments**

The manuscript "Pairing litter decomposition with microbial community structures using the Tea Bag Index (TBI)" combines recently widely applied Tea Bag Index, introduced by Keuskamp et al. (2013), with data on microbial diversity and abundance in soil, green tea and rooibos litter, which should explain the decomposition patterns across three habitats in vineyards. In the whole, the study is rather interesting and highlights the importance of decomposers communities for early stages of decomposition of less or more recalcitrant litter across four seasons. The results are novel, tested hypotheses are clear, the applied methods are relevant, and the results are well discussed and consider the recent literature. The title and the abstract reflect the content of the manuscript. The presentation quality is good.

**Special comments**

I have not detect any serious flows in this study, therefore there are few comments to improve the manuscript.

In introduction it would be good to add some information about stabilization mechanisms, and which factors other than microbes, may influence it, such as links with cations of Al or Ca. Also some notes about the mechanism of colonization of tea bags by bacteria and fungi would be relevant. While fungi proliferate with hyphae, how do bacteria reach tea bags? Is it known, if all the groups of bacteria have the same possibility to get inside the tea bag, so only best decomposers increase their abundance?

Lines 102 – 103: Add the reference to soil types.

Line 202: "The results were surprising…" May be, the lack of correlations of some soil properties with decomposition rate may be caused by small gradients, at least in pH, which is important for microbes activity, and within this study covers only good conditions for decomposition.

Figure 3 has the panels too small and rather difficult to explore. Instead, Figure S3 with ordination results clearly shows the differences between spectra of decomposers, and I'd like to see it in the main text.

**Technical comments**

Lines 116 – 118: The references Tatzber et al. (2015), Keeney (1982) and Kandeler (1993) are absent in the list of references.

Line 133: "were collected at each season for comparison." – This needs clarification. The comparison of microbial community structure?

Line 210: at low temperatures.

Line 224: Table 1 shows.

Lines 266 – 267: sort references by years.

Line 272: "soil green tea and rooibos" – comma after soil.

Also check for the spelling of word "Stabilisation" (as it is usually given in the text) versus "Stabilization" (line 252), the last should be correct. The same about the word "colonization".

---

## Author Response (AR1)

Answers to the editor's comments

**Comments to the author**:
Dear authors,

Two referees were positive about your manuscript and indicated it is publishable after revisions. I would therefore like to invite you to revise your original manuscript according to the suggestions made by the referees and your proposed corrections.

- Thank you for your assessment and for taking the extra time to critically read our manuscript. We are appreciate your comments and hope that the revised version will be acceptable for publication in SOIL.

In addition to both review reports, I have also read your ms with interest and also have a couple of questions/suggestions:
- detail, but it could be mentioned at the end of the introduction (before the hypotheses) that sampling was done over one year. Perhaps it could even be mentioned in the abstract.

- This clarification has now been added to line 99

- Abstract: I found the abstract very clear and a good read, but missed the actual meaning of the results; what does it mean to have a lower S in summer? (apart from the apparent necessity to include colder seasons when determining decomposition dynamics)

- This is a good point and lines 36-38 now read "Stabilisation factor (S), but not decomposition rate (k), correlated with the season and was significantly lower in the summer, indicating a decomposition of a larger fraction of the organic material during the warm months."

- Introduction: like reviewer #1 I think the introduction is somewhat too concise. I miss some background on how different soil types or seasonality might influence the decomposition of recalcitrant or labile plant litter. Also, there seems to be no hypothesis on how soil type or seasonality influence k or S, or microbial diversity? The aims described in the abstract refer to different soil types and seasons.

- Thank you for this comment. We have added a section to the introduction about the influence of seasonality on litter decomposition in lines 107-110. However, since the utilised soil types were in fact highly similar to each other and also the management practices on these soils were identical, we do not feel it right to emphasise this specific topic more. We have included a clarifying sentence regarding expectations of effects of soil type and seasonality to the hypothesis section and hope that the research expectations and hypothesis are now more clear in the revised version.

- Conclusion: this is related to my previous point: in L320 you refer to your hypothesis where you expect seasonality to affect the litter composition rate, but I did not read this specifically in L93-97. It would also be good to read a conclusion about whether the different soil types affected the litter composition rate. I would expect some mention of it after soil types have been mentioned in the aims and is part of the experimental set-up.

- Thank you for this comment. We have now amended the section describing our hypotheses and general scientific expectations (see also  answer to previous point). A conclusion on the effect of soil type is given in lines 325-327. Since we did not observe a major effect, and more importantly since the soil types were in fact highly similar to each other, we prefer to not include a statement on these findings in the conclusions section.

I am looking forward to reading your revised manuscript. Since the reviewers have not responded to your suggested revisions, I will invite both reviewers to look at the revised manuscript as to ask them if they are happy with the changes made.

Kind regards,
Ingrid